# Three-Level Reduced Switch AC/DC/AC Power Conversion System for High Voltage Electric Vehicles

Zahra Sadeghi [1], Mahdi Shahparasti [2,*], Amirhossein Rajaei [3] and Hannu Laaksonen [2]

1   Department of Electrical and Computer Engineering, Isfahan University of Technology,
    Isfahan 84156-83111, Iran; z.sadeghi958@gmail.com
2   School of Technology and Innovations, University of Vaasa, 65200 Vaasa, Finland; hannu.laaksonen@uwasa.fi
3   Department of Electrical and Electronics Engineering, Shiraz University of Technology,
    Shiraz 71557-13876, Iran; a.rajaei@sutech.ac.ir
*   Correspondence:e-mail: mahdi.shahparasti@uwasa.fi

**Abstract:** Two of the main challenges of recent electric vehicles (EVs) are the charging time and high initial cost. To solve the problem associated with long charging time, the car manufacturers are moving from 400 V battery EV (BEV) to 800 V BEV, which enables the utilization of multi-level converters in EV applications. This paper presents a power conversion system consisting of a Vienna rectifier and a two/three level hybrid inverter as a machine-side inverter to drive a permanent-magnet synchronous motor (PMSM). The Vienna rectifier improves the quality of the grid-side current and provides a regulated DC-link voltage. The proposed inverter, known as a 10-switch inverter, offers high output current quality with a lower number of active switches, making it compact and cost-effective. The field-oriented control (FOC), along with the SPWM modulation, is implemented to control the system. A reliable and cost-effective PMSM drive system demands sensorless control; therefore, a sliding mode observer (SMO) is used to estimate the rotor position and velocity. The accuracy of the proposed system was proved through the simulation results from MATLAB/Simulink.

**Keywords:** electric vehicle; fast charging; two/three level inverter; PMSM

## 1. Introduction

Based on sustainable development goals, governments have forced industry to produce more green products to prevent rapid climate change and global warming [1]. Statistics show that near 25 percent of carbon emissions come from internal combustion engine (ICE) vehicles [2,3]. The environmental problems that arise from the ever-increasing consumption of fossil fuels for traction applications, as well as limited natural resources, make a shift in the car industry [4,5]. One viable solution to ease climate change problems is to move from ICE vehicles to electric vehicles, which is a highly effective, but prohibitively costly solution. These days, electric cars are one of the most competitive markets, which new technologies are putting on the brink of change. The most important challenges of electric cars are their high initial cost, long charging time, and the lack of charging stations [6]. To address long charging time, companies such as TESLA are introducing EVs with fast charging capability by increasing charging current. Although this method decreases the waiting time for full battery charge, it causes more heat loss and requires a thicker cable design, resulting in more cost and weight [7,8]. Recently, Porsche has launched fast charging technologies using higher voltage in some luxury and sport EVs. Porsche doubled the charging voltage of luxury EVs because, firstly, using a higher charging voltage can provide fast charging, which significantly reduces the waiting time for full battery charge. Secondly, the higher the charging voltage, the lower the charging current; therefore, the system enjoys a lighter cable design and less heat loss, while fast charging typically leads to heat generation and requires a thicker cable at 400 V EVs [9,10]. Although using a

higher charging voltage brings invaluable benefits, it increases the battery cost and typically requires three-level power electronics structures. Electric cars mainly consist of a battery, an inverter, an electric motor, an onboard charger, and a control unit [11]. The inverter is a key component in EV systems that controls the motor performance [12]. With the emergence of new trends in semiconductors, the two-level half-bridge inverter is the most commonly used converter in low voltage traction applications due to its simple structure and a low number of switching devices [13]. The three-level NPC is introduced as a machine side inverter in the EV drive system [14]. NPC inverters can provide lower current and voltage THD (total harmonic distortion) and less voltage stress across semiconductors than the conventional two-level inverter. Still, the benefits of multi-level converters come at the price of having a higher number of elements. Ref. [15] introduces the Z-source converter as am interface converter in fuel-cell battery hybrid electric vehicles. The Z-source inverter is a single stage power conversion system that has high reliability because of its shoot-through capability, and it also provides a boosted output voltage along with a simple structure. Among different AC machines, permanent-magnet synchronous motors (PMSM) are a potential choice for EVs application because of their high power density and high efficiency. However, the usage of permanent magnets makes PMSMs more expensive than other types of motors [16]. Therefore, sensorless control methods can act as remedies to reduce the total cost of PMSM drive systems. Various control methods have been proposed so far for sensorless control of AC machines. Generally, sensorless control of PMSM is classified into non-adaptive and adaptive control methods. There are three groups of non-adaptive methods, including estimation using phase voltage, phase current, and flux and back-EMF voltage which are reviewed in [17–20]. These methods are quite simple, but they do not have noise-robustness and good performance at low speed. The adaptive methods are divided into two groups: estimation using the model reference adaptive system (MRAS) and state observer estimation. The MRAS technique utilizes a reference model and an adaptive model. Estimation in this method is done by passing the output error of the reference and adaptive model through an adaptive process to bring them closer to each other. Simple structure and easy implementation are the positive features of this method. The disadvantage of this method is its sensitivity to changes in parameters [21,22]. In [23], a first order sliding mode observer (SMO) is designed to estimate the PMSM rotor position. The simple and robust structure of this method has gained considerable attention in sensorless control of AC machines. Still, the chattering problem and the need for phase compensation are the negative points of this method, which are solved in the later version of SMO. The last main part of an EV is the on-board charger. Various DC-DC converter topologies such as buck-boost, Z-source, and resonant converters are proposed for EVs chargers in [24,25]. In [26], the authors present a bridgeless SEPIC converter as a PFC converter in an EV charger. The usage of a bridgeless converter has the potential benefits of lower conduction loss and higher efficiency. High voltage EVs with fast charging capabilities require a cost-effective and reliable charger to enhance the power factor and current quality on the grid-side and provide a regulated DC-link voltage. Ref. [27] reviews several isolated and non-isolated converters that are suitable for the charging infrastructure of EVs. Particularly, multi-level topologies such as the NPC rectifier and Vienna rectifier are presented for fast charging applications because their dual output structure provides a higher voltage and a low voltage stress across switching devices.

In this paper, a new high voltage power conversion system (PCS) is proposed for EV applications to increase system performance on the grid and machine sides and reduce the system size and volume. A two/three level hybrid inverter as presented in [28] is used as a machine-side inverter. This topology uses 10 active switches and produces a combination of three output voltage levels, which provides better voltage and current quality than conventional two-level converters. A Vienna rectifier as a three-level and dual output AC/DC converter is proposed as a battery charger, and a 10-switch inverter is employed to drive a PMSM. A sensorless control scheme is developed to estimate the position and speed of the rotor using the sliding-mode observer method. In summary, the main contributions

of this paper are as follows: (1) proposing a reduced switch AC/DC/AC PMSM drive with three-level outputs; (2) developing a control scheme to control grid current, battery, and PMSM; (3) developing a sensorless controller to reduce the cost of the control system and increase reliability; (4) making an efficiency comparison between the proposed PCS and conventional two-level PCS.

The rest of the paper is organized as follows: in Section 2, the proposed PCS is discussed in terms of inverter topology, modulation method, and losses. Following this discussion, a three-phase three-level Vienna rectifier is described as a battery charger. In Section 3, a sensorless control method is developed using the sliding-mode observer to estimate the rotor's position and speed. The simulation results of the proposed system are presented in Section 4. Finally, Section 5 concludes the study.

## 2. Power Conversion System

### 2.1. Machine Side Converter (10-Switch Inverter)

The power conversion system of the 800 V electric vehicle can be seen in Figure 1. This PCS consists of the Vienna rectifier as a three-phase onboard charger, the 800 V battery packages, the 10-switch inverter, and a PMSM, Figures 2 and 3.

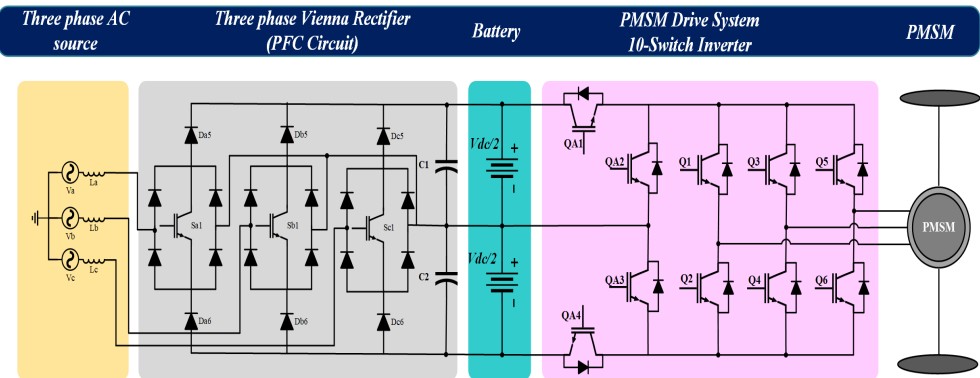

**Figure 1.** Configuration of the proposed power conversion system based on a 10-switch inverter for an 800 V EV system.

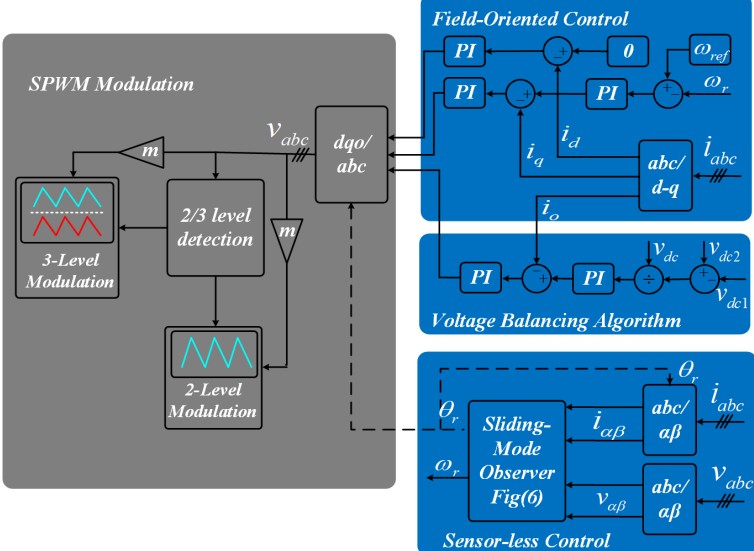

**Figure 2.** The block diagram of the proposed control system for an 800 V EV system with a 10-switch inverter.

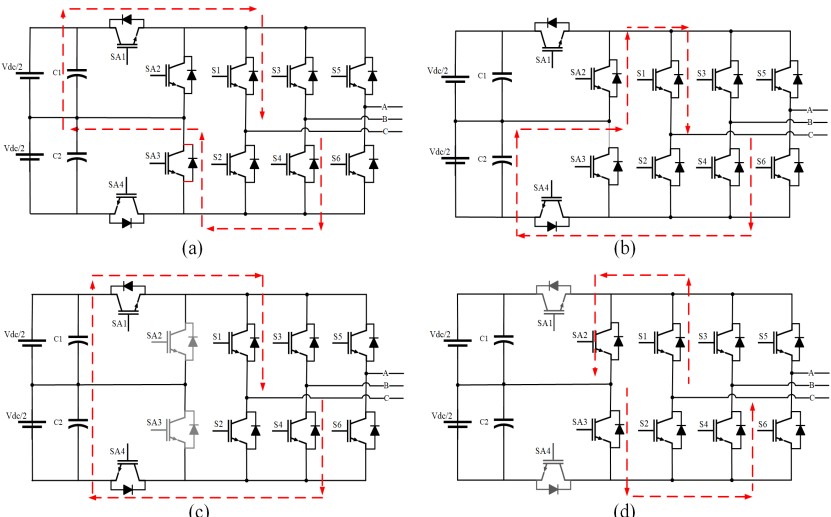

**Figure 3.** 10-switch inverter operation modes in two-level and three-level modes: (**a**) two-level operation (half DC-link voltage), (**b**) two-level operation (half DC-link voltage), (**c**) three-level operation (DC-link voltage), (**d**) zero state operation.

### 2.1.1. Modulation

The 10-switch inverter can be modulated using a combination of three-level and two-level carrier-based modulation techniques. The main procedure of this method was designed based on Figure 4. Considering the particular structure of the inverter, the output voltage can be a combination of two-level or three-level voltages. The main point in controlling the two/three hybrid inverter is to define whether two-level or three-level modulation should be used for each phase. The input signal of the modulation block is a three-phase sinusoidal voltage as formulated in (1), where $V_a^{ref}$, $V_b^{ref}$, and $V_c^{ref}$ are the reference voltages of phases a, b, and, c, and *m* represents the modulation index. In every 60-degree sector, $V_a^{ref}$, $V_b^{ref}$, and $V_c^{ref}$ are defined as the voltages with the minimum ($V_{min}$), medium ($V_{mid}$), and maximum ($V_{max}$) amplitude. Upon detecting the medium voltage, the three-level or two-level modulation is chosen according to the diagram illustrated in Figure 4. In conventional two-level modulation, a single triangular carrier is used, while in the modified three-level modulation, two in-phase carriers are used to modulate the corresponding reference signal [28].

$$
\begin{aligned}
V_a^{ref} &= m \cdot \cos(\omega t) \\
V_b^{ref} &= m \cdot \cos(\omega t - 2\pi/3) \\
V_c^{ref} &= m \cdot \cos(\omega t - 4\pi/3)
\end{aligned}
\tag{1}
$$

$$
\frac{1}{2} + \frac{1}{4} V_{mid} < V_n^{ref} < -\frac{1}{2} + \frac{1}{4} V_{mid}
\tag{2}
$$

Two-level inverters have only eight switching voltage vectors, while the three-level VSI typically presents 27 switching voltage vectors consisting of small, medium, and large vectors, which provide much more controllability in implementing different modulation methods. The characteristics of a 10-switch inverter makes a considerable change in its vector diagram compared to the three-level VSI, restricting inverter performance to producing medium voltage vectors. As indicated in Figure 5, the 10-switch inverter has 21 switching voltage vectors in its space vector diagram. Table 1 depicts the 21 switching vectors, including 12 small vectors, six large ones, and three zero vectors. Consequently, compared to the conventional two-level VSI, the two/three level inverter offers better output voltage quality. Regarding of the lack of seven vectors, the inverter is still anticipated to have almost the same output voltage and current quality compared to the three-level

VSI [29]. Apart from the numerous benefits of the 10-switch inverter, the complex control system could be challenging.

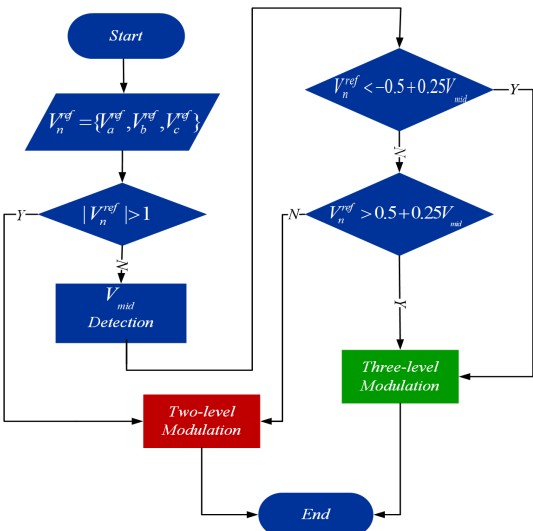

**Figure 4.** The SPWM modulation algorithm for the 10-switch inverter.

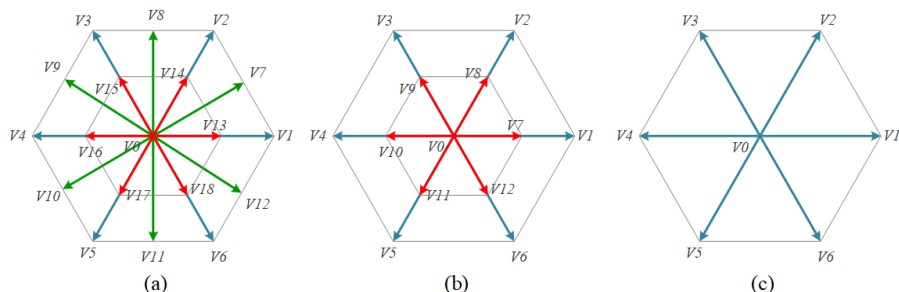

(a)    (b)    (c)

**Figure 5.** Switching vector diagram of different inverters: (**a**) three-level inverter, (**b**) 10-switch inverter, (**c**) two-level VSI.

**Table 1.** Switching vectors of 10-switch inverter.

| Vector Categories | Space Vector | Switching State |
|---|---|---|
| Zero Vectors (0) | $V_0, V_{19}, V_{20}$ | $PPP, NNN, OOO$ |
| Small Vectors ($\frac{V_{dc}}{3}$) | $V_7 – V_{12}$<br>$V_{13} – V_{18}$ | P-type<br>N-type |
| Large Vectors ($2\frac{V_{dc}}{3}$) | $V_1 – V_6$ | $PNN, PPN, NPN$<br>$NPP, NNP, PNP$ |

### 2.1.2. Inverter Losses

The input power of the inverter is expressed as (3), in which $V_{ph}$, $I_{ph}$, and $\varphi$ represent the input phase voltage, input phase current, and the phase difference between voltage and current, respectively. Comparing the conventional 400 V EVs and the proposed system with 800 V DC-link voltage for the same output power, the current in the 10-switch-based system is notably lower (5), resulting in lower power loss and heat.

$$P_{in} = V_{ph} I_{ph} \cos(\varphi) \tag{3}$$

$$\begin{cases} V_{ph,800\text{V}} = 2 V_{ph,400\text{V}} \\ P_{400}^{in} = P_{800}^{in} \end{cases} \tag{4}$$

$$I_{ph,800\text{V}} = \frac{1}{2} I_{ph,400\text{V}} \tag{5}$$

In electrical systems, losses mainly consist of switching loss ($P_{switching}$) and conduction loss ($P_{conduction}$), which define the system's efficiency. Switching power loss is defined by the power loss in turn-on and turn-off commutations. The energy loss for turn-off and turn-on commutation is expressed in (8), where $E_{on}$, $E_{off}$, $v_{ds}$, and $i_d$ denote the turn-on energy loss, turn-off energy loss, drain-source voltage, and drain current, respectively. $t_{i_d}^{10}$ and $t_{v_{ds}}^{10}$ represent the current at 10% of peak value and the voltage at 10% of the $V_{DC}$, respectively [30].

$$P_{loss} = P_{switching} + P_{conduction} \tag{6}$$

$$P_{switching} = P_{on} + P_{off} \tag{7}$$

$$E_{on} = \int_{t_{i_d}^{10}}^{t_{v_{ds}}^{10}} v_{ds}(t) \times i_d(t) dt$$
$$E_{off} = \int_{t_{v_{ds}}^{10}}^{t_{i_d}^{10}} v_{ds}(t) \times i_d(t) dt \tag{8}$$

Since the current in PMSM has a sinusoidal waveform, the total energy loss can be described as a sine wave, with its highest amplitude occurring when the current reaches its peak value [31].

$$E_{sw}(t) = E_{on}(t) + E_{off}(t) = E_{\max} \sin(\omega t) \tag{9}$$

$$P_{sw}^{total} = \frac{1}{T} \int_0^T f_{sw} \times E_{sw}(t) dt = \frac{f_{sw} E_{\max}}{\pi} \tag{10}$$

Some approximation and linearization are required to calculate the conduction loss for a power switch. The conduction loss is formulated in (11); the first term is the average drain current multiplied by the drop voltage of the switch, and the second term defines the power dissipated in equivalent drain-source resistance, where $\overline{i_d}$ and $i_{d,rms}$ represent the average and RMS value of drain current.

$$P_{conduction} = \overline{i_d} v_{ds} + R_{ds(on)} i_{d,rms}^2 \tag{11}$$

### 3. Control Scheme

*3.1. On-Board Charger*

The industry has determined that EV charging systems' interaction with the grid can be divided into three levels regarding charging rates [11]. Therefore, the three-phase Vienna rectifier can be categorized as Level II, with a power level up to 20 kW. The three-level three-phase Vienna rectifier with a simple structure offers boosted output voltage and improved current quality on the grid side. It is a well-known three-level rectifier with a reduced switch count that has been widely used in high-power charging systems. The main advantages of this converter are as follows:

- Low voltage stress across switches;
- Low number of active switches and high efficiency;
- Ability to balance output voltages and control the neutral point voltage.

This converter links the three-phase AC supply to the 10-switch inverter with a maximum DC voltage of up to 800 V, making it a suitable choice for an on-board charger in the proposed 800 V EV system. Generally, power factor correction in three-phase systems can be accomplished using at least three active switches and a suitable control system. Several control strategies have been presented in the literature to provide the required power and unity power factor operation, including hysteresis current control, vector control, and model predictive control. In this study, the power factor enhancement is done using vector control in the direct-quadrature ($d - q$) reference frame. The objective of this method is to control the $d$ and $q$ components of injected current to control active and reactive power. The diagram of the grid-side control method is shown in Figure 6. It contains two

control loops to regulate DC-link voltage (charge batteries) and grid-side power. In the $i_d$ control channel, $I_{bat}$ is defined based on the nominal current of the battery to implement the constant-current constant-voltage (CC-CV) charging control algorithm. The reference current in the $q$ reference frame is defined based on a reactive power setpoint equal to zero for unity power factor operation. When voltage balancing of two output capacitors is required, it can be achieved simply by adding a DC component to the reference currents, as illustrated in Figure 6 [32].

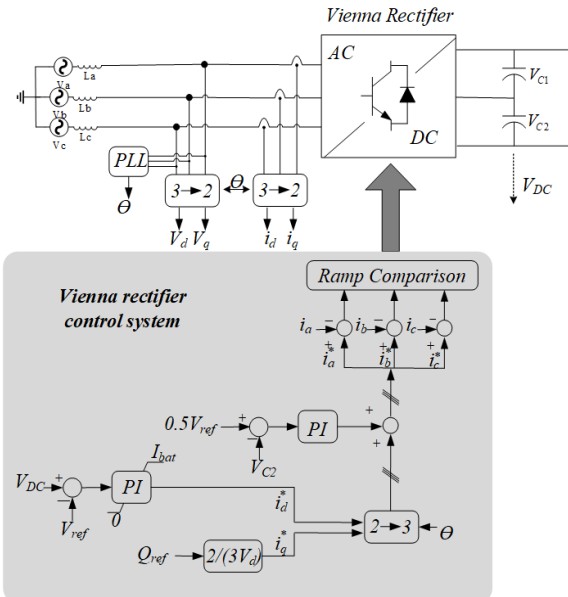

**Figure 6.** The block diagram of the Vienna rectifier control system.

### 3.2. PMSM Model

PMSMs with high power density, reliability, efficiency, and wide-speed operation are used in various industries, including electric vehicle applications. Sensorless drive systems typically offer higher reliability, lower initial cost, and lower maintenance costs. Since the cost of EVs plays a considerable role in the market, sensorless control of the PMSM drive seems to be a solution to slightly reduce the drive system's cost. The following equations in (12) and (13) display the PMSM dynamic model in the fixed reference frame, which is helpful for the implementation of the sensorless control algorithm in next section. $v_{\alpha,\beta}$ and $i_{\alpha,\beta}$ are the voltage and current, respectively, in the $\alpha - \beta$ frame, and $\omega_r$ $\theta_r$, $\varphi_{pm}$, $r$, and $L$ represent the rotor speed, rotor position, permanent magnet flux, and the stator equivalent resistor and inductance.

$$\begin{bmatrix} v_\alpha \\ v_\beta \end{bmatrix} = (r + L\frac{d}{dt})\begin{bmatrix} i_\alpha \\ i_\beta \end{bmatrix} + \omega_r.\varphi_{pm}\begin{bmatrix} -\sin(\theta_r) \\ \cos(\theta_r) \end{bmatrix} \tag{12}$$

$$\begin{bmatrix} \varphi_\alpha \\ \varphi_\beta \end{bmatrix} = L\begin{bmatrix} i_\alpha \\ i_\beta \end{bmatrix} + \varphi_{pm}\begin{bmatrix} \cos(\theta_r) \\ \sin(\theta_r) \end{bmatrix} \tag{13}$$

The well-known FOC method is frequently used in PMSM control systems because of its good transient and steady-state response, constant switching frequency, and low torque ripple. In this method, the motor equations are transferred to the rotor reference frame formulated in (14) and (15), where $v_{d,q}$, $i_{d,q}$ are the voltage and current, respectively, in the $d - q$ frame, and $\omega_r$ and $\varphi_{pm}$ represent the rotor speed and permanent magnet flux. In a rotating reference frame, the stator current component of the first axis is proportional to the flux linkage, and the current component of the other axis is proportional to the torque. As in the FOC method diagram depicted in Figure 2, the reference speed is compared to the measured speed, and the error is delivered to the speed controller. The output of this

controller is the reference value of $i_q$, and for maximum torque achievement, the reference value of $i_d$ is considered zero.

$$\begin{bmatrix} v_d \\ v_q \end{bmatrix} = \begin{bmatrix} (r + L\frac{d}{dt}) & -L\omega_r \\ L\omega_r & (r + L\frac{d}{dt}) \end{bmatrix} \begin{bmatrix} i_d \\ i_q \end{bmatrix} + \omega_r \begin{bmatrix} 0 \\ \varphi_{pm} \end{bmatrix} \tag{14}$$

$$\begin{bmatrix} \varphi_d \\ \varphi_q \end{bmatrix} = L \begin{bmatrix} i_d \\ i_q \end{bmatrix} + \begin{bmatrix} \varphi_{pm} \\ 0 \end{bmatrix} \tag{15}$$

### 3.3. Sensorless Control Method

Using mechanical sensors reduces the system's reliability. The failure of a mechanical sensor generally disrupts the whole system's performance, since the vector control of PMSM is entirely dependent on the rotor position. In addition, mechanical sensors require complex wiring, control systems, and maintenance, making them less reliable than electrical systems. In this section, a robust and reliable sensorless method using a sliding mode observer (SMO) is designed to estimate the rotor position and velocity in PMSM. Sliding mode control is a nonlinear method that provides high accuracy and robustness and relatively simple implementation. It is commonly used for systems with non-precise models and high external perturbations. The equations in (16) describe the PMSM current equation in the stator reference frame, where $i_{\alpha\beta}$, $\dot{i}_{\alpha\beta}$, $v_{\alpha\beta}$, and $e_{\alpha\beta}$ represent the stator current and the first derivative of the current, stator voltage, and back-EMF in the $\alpha\beta$ reference frame, respectively. The state equation for the estimated current is shown in (18), where $\hat{i}_{\alpha,\beta}$ and $\hat{r}$ denote the corresponding estimated variable. $\hat{z}_{\alpha,\beta}$ is the estimated back-EMF, which is obtained by passing the error between the estimated and measured current through the Sigmoid function defined in (20), in which $k$ is the observer gain and $a$ is the coefficient to adjust the function slop [33].

$$\dot{i}_\alpha = \frac{r}{L} i_\alpha - \frac{1}{L} e_\alpha + \frac{1}{L} v_\alpha \tag{16}$$

$$\dot{i}_\beta = \frac{r}{L} i_\beta - \frac{1}{L} e_\beta + \frac{1}{L} v_\beta \tag{17}$$

$$\dot{\hat{i}}_\alpha = \frac{\hat{r}}{L} \hat{i}_\alpha + \frac{1}{L} v_\alpha - \frac{1}{L} \hat{z}_\alpha \tag{18}$$

$$\dot{\hat{i}}_\beta = \frac{\hat{r}}{L} \hat{i}_\beta + \frac{1}{L} v_\beta - \frac{1}{L} \hat{z}_\beta \tag{19}$$

$$\begin{bmatrix} \hat{z}_\alpha \\ \hat{z}_\beta \end{bmatrix} = \begin{bmatrix} k \cdot H(\hat{i}_\alpha - i_\alpha) \\ k \cdot H(\hat{i}_\beta - i_\beta) \end{bmatrix} = k \begin{bmatrix} (\frac{2}{1+\exp(-a \cdot (\hat{i}_\alpha - i_\alpha))}) - 1 \\ (\frac{2}{1+\exp(-a \cdot (\hat{i}_\beta - i_\beta))}) - 1 \end{bmatrix} \tag{20}$$

$$\hat{r} = \frac{1}{L} [(\hat{i}_\alpha - i_\alpha) \cdot \hat{i}_\alpha + (\hat{i}_\beta - i_\beta) \cdot \hat{i}_\beta] \tag{21}$$

The rotor position estimation is done by defining a sliding surface, described in (22) as the error between the estimated current and the measured one. The controller forces the state variable to move towards the sliding surface by restricting movement on the sliding surface, moving the surfaces towards the equilibrium state. Therefore, the error between the estimated and measured current becomes zero, as the sliding surface is zero in a stable system. Figure 7 shows the diagram of the sensorless control of the PMSM, which consists of the SMO, filter block, and the estimation block, including a phase-locked-loop.

$$S_n = \begin{bmatrix} S_\alpha \\ S_\beta \end{bmatrix} = \begin{bmatrix} \hat{i}_\alpha - i_\alpha \\ \hat{i}_\beta - i_\beta \end{bmatrix} \tag{22}$$

By ignoring the resistance estimation part and defining $\bar{i}_\alpha = \hat{i}_\alpha - i_\alpha$ and $\bar{i}_\beta = \hat{i}_\beta - i_\beta$, the Lyapunov function for investigating the sliding condition of PMSM sensorless control is as follows:

$$V = \frac{1}{2} S_n^T S_n \ , \quad \dot{V} = S_n^T \dot{S}_n < 0 \tag{23}$$

As has been proven in previous studies, system stability is achieved if the sliding surface satisfies the condition in (25), which declares that the observer gain must meet the following requirement in (26):

$$\dot{V} = \begin{bmatrix} \bar{i}_\alpha & \bar{i}_\beta \end{bmatrix} \begin{bmatrix} \frac{r}{L}(\hat{i}_\alpha - i_\alpha) + \frac{1}{L}(\hat{e}_\alpha - \hat{z}_\alpha) \\ \frac{r}{L}(\hat{i}_\beta - i_\beta) + \frac{1}{L}(\hat{e}_\beta - \hat{z}_\beta) \end{bmatrix} \tag{24}$$

$$\dot{V} = -\frac{r}{L}(\bar{i}_\alpha^2 - \bar{i}_\beta^2) + \frac{\bar{i}_\alpha}{L}(\hat{e}_\alpha - kH(\bar{i}_\alpha)) + \frac{\bar{i}_\beta}{L}(\hat{e}_\beta - kH(\bar{i}_\beta)) < 0 \tag{25}$$

$$k \geq \max(|\hat{e}_\alpha|, |\hat{e}_\beta|) \tag{26}$$

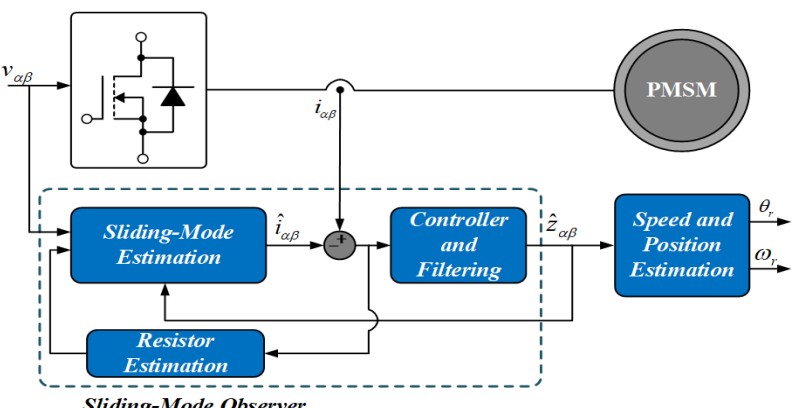

**Figure 7.** Diagram of the sensorless control of the PMSM using SMO (part of Figure 2).

## 4. Simulation Results

This section depicts the simulation results of the proposed drive system for a fast-charging electric vehicle from MATLAB/Simulink. The simulation results were obtained for three modes, including the motoring operation of the PMSM, regenerative braking, and static charging operation in which the inverter and the PMSM are not working. The presence of the Vienna rectifier on the grid-side as an on-board charger connects the system directly to the mains. The batteries are supplied by a DC-link voltage of up to 800 V. Then, the battery packs are connected to a two/three level inverter that feeds the PMSM. As mentioned earlier, one of the major obstacles to adopting electric vehicles worldwide is the long battery charging time. Therefore, new vehicles charged by higher voltages can significantly reduce the batteries' full charge time. Table 2 presents the characteristics of the proposed EV drive system. The simulation results compare an 800 V system with a 10-switch inverter and a 400 V system with the VSI. In the following, the simulation results of the proposed system are described in detail to validate the statements claimed in this paper. The 10-switch inverter can simultaneously operate in two-level and three-level modes. In other words, one phase can be modulated by three-level modulation, and the two other phases may be modulated as two-level. Figures 8 and 9 compare the line-to-line voltage and the common-mode voltage of the 10-switch inverter and the standard two-level VSI at the PMSM-rated speed of 2500 r/min, respectively. It can be seen that the line-to-line voltage waveforms of the 10-switch inverter have three levels of voltage with higher quality.

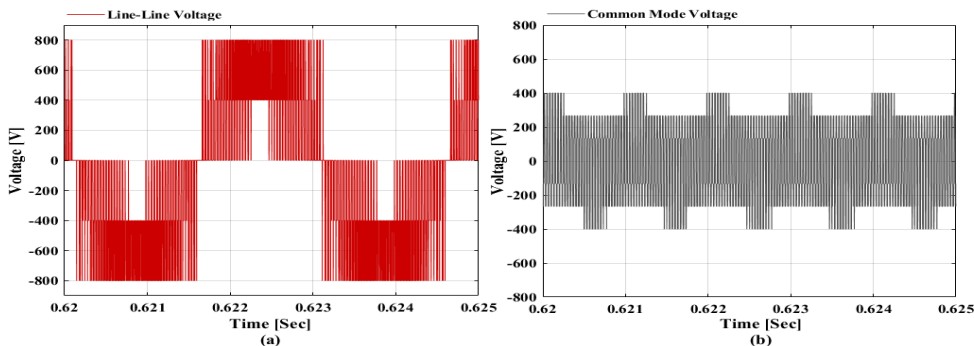

**Figure 8.** Simulation result of the 10-switch inverter with 800 V DC voltage at 2500 r/min: (**a**) line-to-line voltage, (**b**) common mode voltage.

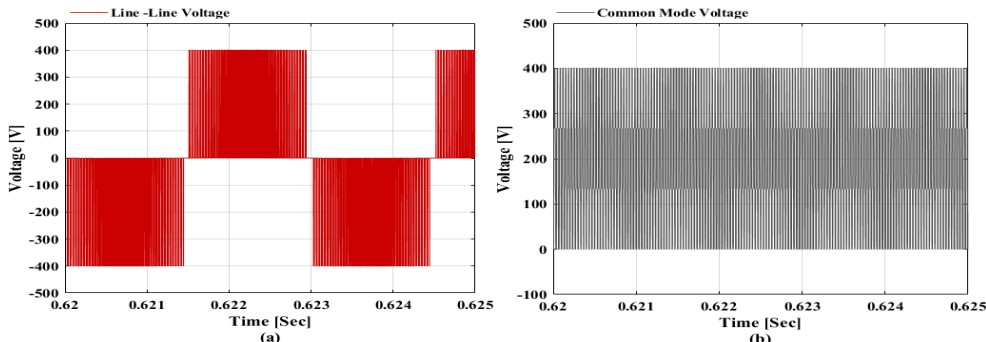

**Figure 9.** Simulation results of the two-level VSI with 400 V DC voltage at 2500 r/min: (**a**) line-to-line voltage, (**b**) common mode voltage.

**Table 2.** Characteristics of the proposed drive system.

| PMSM Parameter | Value | Control Parameter | Value |
|---|---|---|---|
| PMSM Rated Speed | 2500 r/min | AC Input Voltage(rms) | 380 V |
| PMSM Rated Torque | 100 N·m | DC-link Voltage | 800 V |
| Rated Power | 26 kW | Switching Frequency | 30 KHz |
| Stator Inductance/Resistance | 1 mH/0.25 Ω | k (SMO Gain) | 2000 |
| Number of Poles | 8 | a (Sigmoid Constant) | 0.054 |

The DC-link voltage in the proposed drive system is twice that in the standard VSI. The PMSMs used in this simulation are compatible with their associated systems, making way for a lower current amplitude and lower loss in the proposed system based on a 10-switch inverter. Table 3 represents the parameters of two PMSMs used in this study. The 800 V system is designed based on a PMSM with higher flux linkage and higher voltage, resulting in a lower current to provide the same output power as the 400 V system. Figure 10a,b compares the PMSM current in the proposed 800 V EV system and the conventional EV system with a 400 V battery pack for the same output power. Section 2 described how to calculate the switching and conduction losses in the proposed 800 V inverter and the 400 V VSI for nominal load. In Table 4, specific SIC MOSFETs are selected for this simulation based on the system's power rating and the switches' blocking voltage. Although the number of elements in the 10-switch inverter is higher than the VSI, the significant reduction in current results in lower switching and conduction losses in the proposed system compared to the conventional system shown in Figure 11a. Figure 11b depicts the efficiency curve for the 400 V VSI and the 800 V 10-switch over a wide range of load variation, which proves that due to the fact the 800 V system has fewer losses, it offers higher efficiency. The simulation

result of the input current and output voltage of the Vienna rectifier as an on-board charger are illustrated in Figure 12a,b.

**Table 3.** Characteristic PMSM parameters for the 400 V and 800 V systems.

| PMSM Parameter | 400 V System | 800 V System |
|---|---|---|
| Rated power (kW) | 30 | 30 |
| Rated voltage (V) | 250–450 | 450–600 |
| Max current (A) | 150 | 95 |
| Max torque (N·m) | up to 150 | up to 100 |
| Max speed (r/min) | up to 3000 | up to 6000 |
| Flux linkage (mWb) | 45 | 98 |
| Moment of inertia (kg·m$^2$) | 0.08 | 0.08 |

**Table 4.** Specification of selected SIC switches in the 400 V VSI and 800 V 10-switch inverter.

| Inverter Topology | Leg | Part Number | Blocking Voltage | Continuous Current Rating | Rds,on |
|---|---|---|---|---|---|
| VSI | All three legs | C3M0025065D | 650 (V) | 97 (A) | 25 (mΩ) |
| 10-Switch | The leg with four series switches | C3M0015065D | 650 (V) | 81 (A) | 15 (mΩ) |
| 10-Switch | Legs with two series switches | C3M0021120D | 1200 (V) | 81 (A) | 21 (mΩ) |

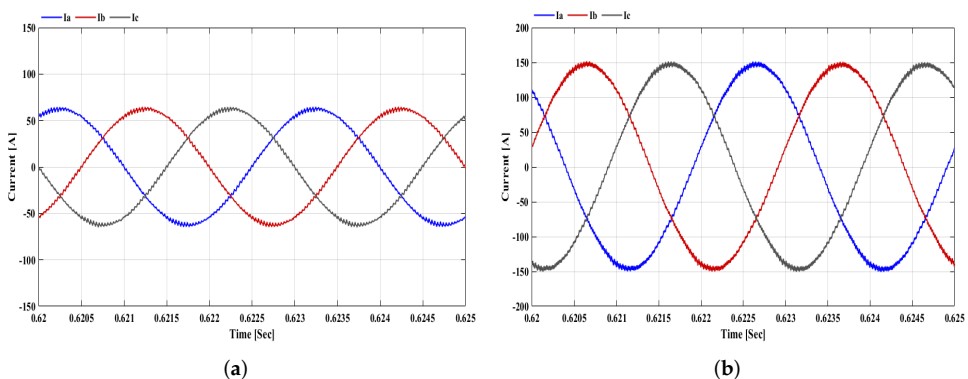

(a)

(b)

**Figure 10.** Simulation results of PMSM phases current at 2500 r/min: (**a**) 10-switch inverter, (**b**) two-level VSI.

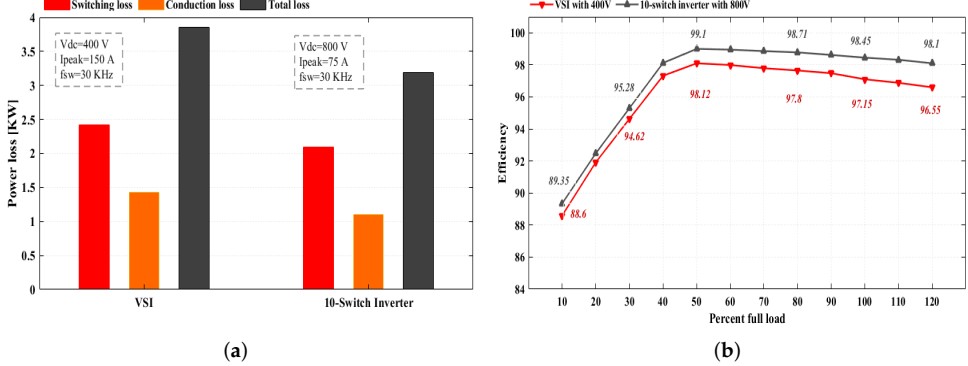

(a)

(b)

**Figure 11.** Comparison of the 10-switch inverter with 800 V DC voltage and the two-level VSI with 400 V DC voltage in terms of (**a**) power loss for full load operation and (**b**) efficiency for a wide range of load current.

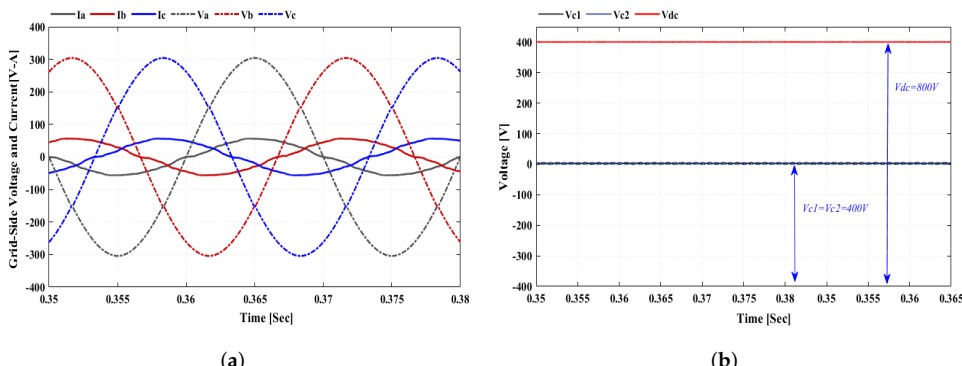

**Figure 12.** Simulation results for the Vienna rectifier: (**a**) the input current of the charger, (**b**) the output voltage of the charger.

Generally, a PMSM works under two main operation states shown in Figure 13: constant torque for speed variations up to the rated speed, and the constant power region for speed above nominal. In this study, system performance is examined at a nominal speed of 2500 r/min. Figure 14a demonstrates the system speed response for accelerating, constant speed, and regenerative braking operation. The output torque waveform with torque characteristic of the brake pedal is shown in Figure 14b to investigate the system's response to torque changes in different operation regions. The system torque and speed response results reveal that the proposed EV drive can adequately track the torque and speed changes in the accelerating and braking regions. To show the effect of normal motoring operation and regenerative braking on the state of charge (SOC) of the batteries, the batteries' SOCs for both systems with 800 V and 400 V battery packs are reported in Figure 15a,b. Since the proposed system works with a higher voltage, the current amplitude of its battery is notably lower than the current amplitude in the standard VSI, leading to a thinner cable design and lower heat loss. Finally, the results of implementing an improved method of conventional SMO for estimating the rotor position are presented. This method calculates the back-EMF voltage using the Sigmoid function, which is a continuous sign function model. The error between the measured current and the estimated current is negligible in Figure 16a,b, which proves that the rotor position is estimated correctly. As shown Figure 16c, the estimated rotor position and the measured one are well matched.

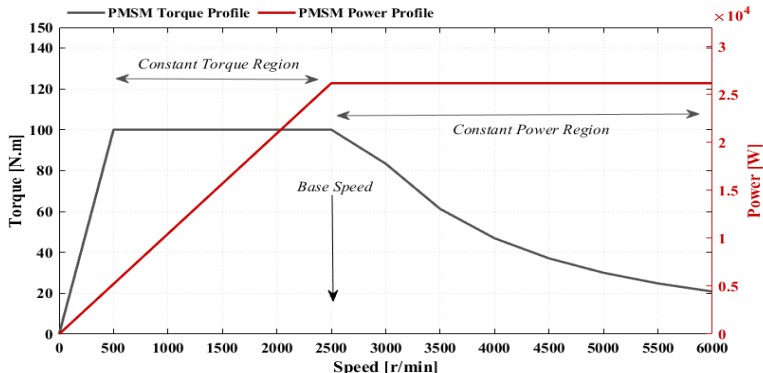

**Figure 13.** PMSM torque and power profile over a wide range of speeds.

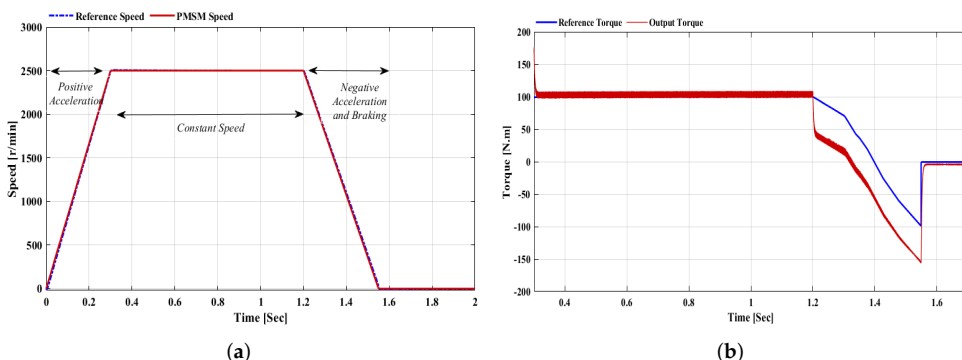

**Figure 14.** Simulation results for the PMSM: (**a**) speed response in normal motoring operation and regenerative braking, (**b**) torque response in normal motoring operation and regenerative braking.

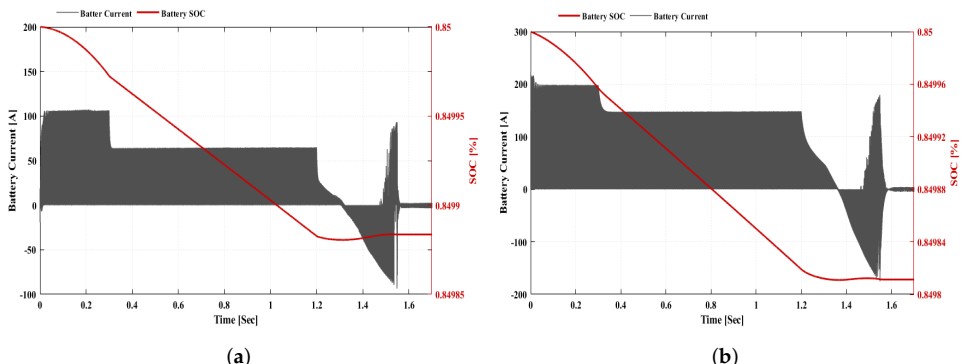

**Figure 15.** Simulation results of battery current and SOC: (**a**) 10-switch inverter, (**b**) two-level VSI.

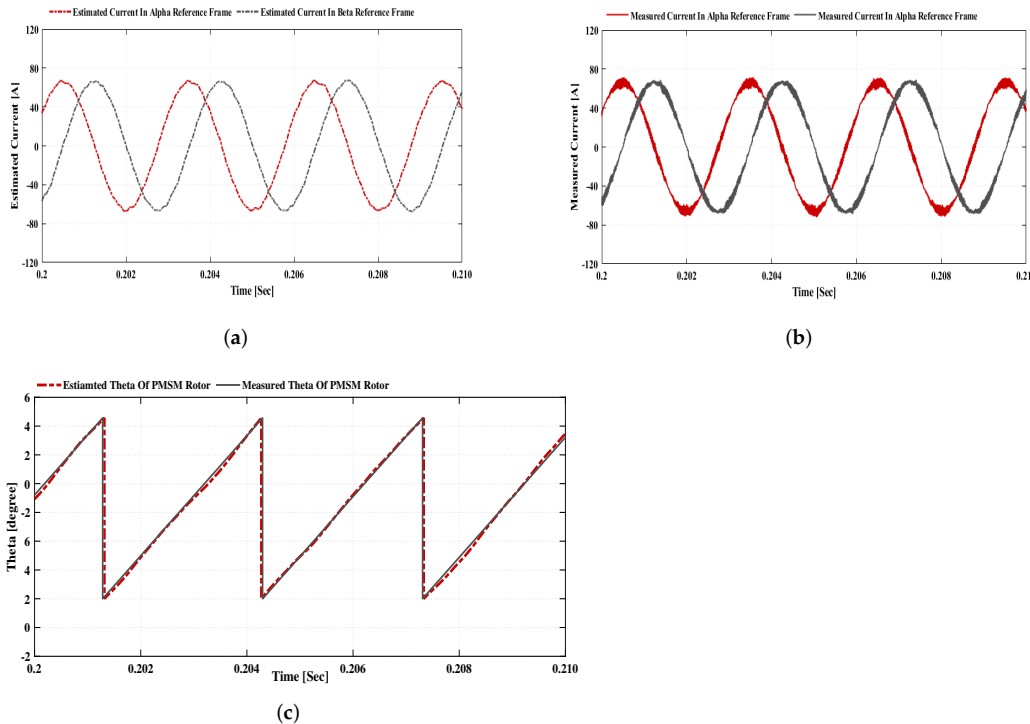

**Figure 16.** Simulation results: (**a**) PMSM estimated current in alpha-beta reference frame, (**b**) PMSM measured current in alpha-beta reference frame, (**c**) comparison of estimated and measured rotor position.

## 5. Conclusions

This paper proposed an 800 V PCS for EV applications using a 10-switch inverter. The utilization of doubled voltage in the proposed EV system brings significant benefits, including reduced charging time, enhanced efficiency due to the reduced switching and conduction loss, and lighter cable design. In the 800 V EV system, three-level converters are much more competitive compared to two-level VSIs. Therefore, the 10-switch is proposed as a machine-side inverter, possessing fewer switching elements than other three-level inverters. The FOC control and hybrid two-three level SPWM control the 10-switch inverter. The switching vectors of 10-switched converters were investigated, showing that the two/three level hybrid inverter should work under a combination of two-level and three-level modulation. It has been illustrated that the 10-switch inverter offers lower voltage and current THD than a standard two-level VSI. A comparison of the proposed 800 V system and the two-level VSI has been made, describing the superior features of the proposed method in efficiency and output current quality. Additionally, a sensorless control algorithm known as a high-speed SMO was formulated and designed to estimate the PMSM rotor position, which results in enhanced system reliability and reduced system cost. The simulation results of the proposed PCS from MATLAB/Simulink validated the adequate performance of the proposed system.

**Author Contributions:** Conceptualization, Z.S. and M.S.; methodology, Z.S. and M.S.; validation, Z.S., M.S., and A.R.; formal analysis, Z.S., M.S., A.R., and H.L.; writing—original draft preparation, Z.S.; review and editing, M.S., A.R., and H.L. All authors have read and agreed to the published version of the manuscript.

**Funding:** This research received no external funding.

**Institutional Review Board Statement:** Not applicable.

**Informed Consent Statement:** Not applicable.

**Data Availability Statement:** Data available on request due to privacy restrictions.

**Conflicts of Interest:** The authors declare no conflict of interest.

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
