# Peer review of "Three-Level Reduced Switch AC/DC/AC Power Conversion System for High Voltage Electric Vehicles"

_sustainability, doi:10.3390/su14031620_

Round 1
Reviewer 1 Report
Dear authors
The paper presents a Three-level Reduced Switch AC/DC/AC Power Conversion System for High Voltage Electric Vehicles. However, there are some comments for the authors. The comments are:
- Usually, EV power conversion systems start with the battery as a source. How can the EV source be a three-phase?! Hence the rectifier will be useless for the system. If you decide to propose the charging station, the PMSG will not be operated during the charging process. I think that the proposed system is not suitable for EV power systems.
- The (Ten-Switch Three-Phase Three-Level Inverter) and Vienna rectifier are well-known converters in the literature. Hence, where is the novelty and contribution of the paper?
Author Response
Dear Reviewer,
The authors would like to appreciate your effort in reviewing our paper and raising valuable comments. The manuscript has been revised according to the comments of the reviewers. All the reviewer's comments are addressed individually and the manuscript has been revised, carefully.
Yours sincerely,
Authors
The paper presents a Three-level Reduced Switch AC/DC/AC Power Conversion System for High Voltage Electric Vehicles. However, there are some comments for the authors. The comments are:
Comment 1) Usually, EV power conversion systems start with the battery as a source. How can the EV source be a three-phase?! Hence the rectifier will be useless for the system. If you decide to propose the charging station, the PMSG will not be operated during the charging process. I think that the proposed system is not suitable for EV power systems.
Answer 1) An EV system mainly consists of an onboard charger, battery-package, inverter, and PMSM. The three-phase Vienna rectifier is used as an onboard charger (the commercialized three-phase onboard chargers are now available in the market) and 10-switch inverter is proposed to drive PMSM. We represented the whole conversion system of the proposed 800V EV in one single figure to give the reader a better understanding of the 800V battery using a three-level Vienna rectifier as the onboard charger. It should be mentioned that showing all aspects together does not mean that all parts are working simultaneously. For sure, during the charging intervals, the PMSM is not working, and on the other hand, during PMSM operation, the Vienna rectifier is not connected to the grid.
Comment 2) The (Ten-Switch Three-Phase Three-Level Inverter) and Vienna rectifier are well-known converters in the literature. Hence, where is the novelty and contribution of the paper?
Answer 2) Thank you very much for your comment. The new 800V EVs continue to attract more and more attention nowadays. When it comes to 800V systems, the 3-level converters seem to be much more competitive compared to 2-level VSI in terms of efficiency and loss. The most apparent short shortcoming of such 3-level converters as NPC is the high number of switching devices. So, the novelty of this paper lies in the novel application of the reduced switched 10-switch converter for upcoming 800V EVs. The number of papers about 10-switch converter is limited to less 10 papers in total based on our googling and searching in different journals, and this paper is the first paper, which wants to use 10 switch converter in the field of EV.
Reviewer 2 Report
Recommendation: minor revision.
In this manuscript, the authors proposed a new 800V high voltage power conversion system for electric vesicles application using PMSM fed by a 10-switch inverter. The manuscript with good logical, well-organized and the accuracy of the proposed system is proved through the simulation results from MATLAB/Simulink. The experiment data and explanations offered in this manuscript are sufficient to support the conclusion. Therefore, I recommend that this manuscript can be accepted for Sustainability after minor revision. Some issues that need to be further settled are as follow:
- There too many Figures in the manuscript, some Figures should be combined and
- The authors compared the electrochemical performance by using the 10-switch inverter with 800V DC voltage and Two-level VSIwith 400V DC voltage. If can provide some photos would be better.
- The detailed information of the equipment and the tested parameter batter are offered in the manuscript.
- These operating parameters of 800V high voltage power conversion system are accepted by the electric vesicles? Can you provide a fast charging video for electric vesicles?
Author Response
Dear Reviewer,
The authors would like to appreciate your effort in reviewing our paper and raising valuable comments. The manuscript has been revised according to the comments of the reviewers. All the reviewer's comments are addressed individually and the manuscript has been revised, carefully.
Yours sincerely,
Authors

Reviewer 3 Report
Title : Three-level Reduced Switch AC/DC/AC Power Conversion System for High Voltage Electric Vehicles
Authors : Zahra Sadeghi, Mahdi Shahparasti, Amirhossein Rajaei and Hannu Laaksonen.
-----------------------
In this paper, the authors present a new high voltage power conversion system for electric vehicles applications, to increase the system performance on the grid and machine sides and reduce the system size and volume.
Strengths:
. The abstract presents correctly and synthetically the paper.
. The question set in this paper is original and well defined.
. The bibliography part has been worked, and is well analysed in the introduction section.
. Section 2 is very interesting and illustrated.
Weaknesses:
. Section 3 could be more precise, especially at the modelling level.
. Some limits of the results obtained in this paper are mentioned.
. Conclusion could also gain in being extended.
Significance:
. The scientific content of this paper is correct for me and deserves to be published.
. The hypotheses are correctly identified as such, and the presented results are appropriately presented and significant.
. The technical quality of this paper is correct for me.
. The limits of the results obtained in this paper are not mentioned. This point could be investigated.
. Anyway, I took interest and pleasure to read this paper.
Quality of presentation:
. The abstract is clear and presents correctly the subject addressed in this paper.
. This paper contains the basic sections of a scientific paper.
. The subheadings used for the redaction of this paper make it clear.
. This paper is clear, easy to follow and to read, and logically written.
. The data and analyses are appropriately presented.
. Conclusion could gain in being extended.
Scientific soundness:
. The subject addressed in this paper is relevant. The study has been correctly designed, and is technically sound.
. The methods and software could be more deeply described, with enough details to allow another researcher to reproduce the results.
. In my opinion, the conclusions of this paper are interesting, but should be extended and highlighted for the readership of the journal.
Overall evaluation:
. The English language quality and style of this paper are appropriate and understandable.
. I think there is an overall benefit to publish this work.
As a conclusion, my suggestion to the editor is to accept this paper for publication in Sustainability.
References :
--------------
. 32 research references, with no non-research reference, out of which 1 self-reference, giving a self-reference good ratio equal to 3%.
. The bibliography of this paper is mainly composed of recent references: 8 of them are more than 10 years old, and 24 of them are less than 10 years old.
. Please avoid the formulation 'et al.' in the references section: ([7]). The complete list of authors deserve to be cited in this section.
Typos / Comments / Remarks:
------------------------------------
. Line 1: Vesicles --> Vehicles
. Line 17: Statics --> Statistics
. Lines 95-96: The remainder does not correspond to the actual organization of the paper.
. Line 237: Fig 19 is zero ?
. Figure 19: alpha reference frame --> beta reference frame.
Author Response

(The authors gave the same response as above.)

Round 2
Reviewer 1 Report
Comments for authors:
- In lines 220-222, you said that both Fig 10(a) and (b) have the same output power at 2500rpm (ie the same torque). As the motor is PMSM the air gap flux is constant. Hence, the motor torque is proportional to the motor current. However, the motor current in (b) has become three times that of (a). How can the power be the same?! I think that there is some error.
- In Fig. 16, the SOC is not synchronized with the battery current, Why?
- The SOC curves of Fig 16. are different from Fig. 17, How?
- What is the rated current of the PMSM?
Author Response
Dear Reviewer,
The authors would like to appreciate your effort in reviewing our paper and raising valuable comments.
The answer to your comments is attached.
Yours sincerely,
Authors

Round 3
Reviewer 1 Report
No comments